# Probing electrochemical reactions in organic cathode materials via in operando infrared spectroscopy

Alen Vizintin [1], Jan Bitenc[1], Anja Kopač Lautar[1], Klemen Pirnat[1], Jože Grdadolnik[1], Jernej Stare[1], Anna Randon-Vitanova[2] & Robert Dominko [1]

Organic materials are receiving an increasing amount of attention as electrode materials for future post lithium-ion batteries due to their versatility and sustainability. However, their electrochemical reaction mechanism has seldom been investigated. This is a direct consequence of a lack of straightforward and broadly available analytical techniques. Herein, a straightforward in operando attenuated total reflectance infrared spectroscopy method is developed that allows visualization of changes of all infrared active bands that occur as a consequence of reduction/oxidation processes. In operando infrared spectroscopy is applied to the analysis of three different organic polymer materials in lithium batteries. Moreover, this in operando method is further extended to investigation of redox reaction mechanism of poly (anthraquinonyl sulfide) in a magnesium battery, where a reduction of carbonyl bond is demonstrated as a mechanism of electrochemical activity. Conclusions done by the in operando results are complemented by synthesis of model compound and density functional theory calculation of infrared spectra.

[1] National Institute of Chemistry, Hajdrihova 19, 1000 Ljubljana, Slovenia. [2] Honda R&D Europe GmbH, Carl-Legien Strasse 30, 63703 Offenbach, Germany. A. Vizintin and J. Bitenc contributed equally to this work. Correspondence and requests for materials should be addressed to R.D. (email: robert.dominko@ki.si)

# ARTICLE

A lthough Li-ion batteries are the most suitable mature battery technology for commercial applications, researchers are actively exploring the alternatives like Li–S, Li–O₂, Na-ion, Mg batteries and different metal–organic batteries[1–4]. Among these technologies metal–organic batteries are attracting increasing attention due to their versatility, low cost and sustainability[5,6]. Unlike most inorganic materials, organic materials can be used with a variety of counter ions, which removes any possible concerns about Li supply out of the picture[7]. Due to the electrochemical conversion reaction, they bypass the intercalation and solid-state diffusion limitations inside inorganic hosts and can achieve high rates without the need for nanosizing the materials. Use of organic cathode materials has already been successfully demonstrated for Li[8–10], Na[11,12], Mg[13–15] and K[16] battery systems. Although some of the organic cathode materials show stable and long-term cycling, the electrochemical mechanism in certain materials is still under debate. Most often electrochemical mechanism of organic materials is presumed only on predictions and analogies based on post-mortem measurements. The main reason is lack of analytical techniques that allow simultaneous electrochemical and spectroscopic characterization (i.e., in operando investigations) and rather fast degradation of ex situ samples coupled with troublesome handling, especially in their discharged state. As a result, the proposed electrochemical reaction mechanisms are rarely investigated, much less confirmed.

So far, the electrochemical reaction mechanism inside of organic battery cathodes have been investigated by X-ray diffraction[17], nuclear magnetic resonance (NMR)[18], Raman and infrared (IR) spectroscopy[17,19,20]. However, there are certain limitations to these techniques; X-ray diffraction is limited to crystalline samples and Raman spectroscopy is troublesome due to fluorescence and laser-induced sample damage. Both IR and NMR techniques have been so far limited to the analysis of ex situ samples[20,21]. In this context development of in operando techniques is needed for future progress of both metal–organic batteries and other types of batteries. In operando techniques help us improve our knowledge about mechanisms of battery operation and degradation and in future might allow real-time monitoring of the battery state of health[22].

Herein we show in operando measurements based on attenuated total reflectance (ATR)-IR of the electrode composite inside a modified pouch cell on the established redox-active polymer, poly(anthraquinonyl sulfide) (PAQS), that has been used in several battery systems (Li, Na, Mg, K)[8,14,16,23]. Measurements are performed in Li and Mg battery systems. We employ subtractive normalization of the obtained IR spectra for visualization of the IR-active spectral changes caused by electrochemical cycling of the electrode. The reduction of the carbonyl bond in Li and Mg PAQS systems is confirmed. Good reversibility in both battery systems confirms the applicability of organic electrodes as low cost, versatile and sustainable cathode materials. In operando measurements are complemented by comparing IR spectra of a simple anthraquinone (AQ) and a synthesized lithium salt of 9,10-dihydroxy anthracene (Li₂AQ). To confirm the applicability of our in operando ATR-IR method to other compounds, we have extended our study to polyanthraquinone (PAQ) and polyaniline (PANI) electroactive compounds. Furthermore, interpretation of experimental ATR-IR spectra is supported by quantum mechanical calculation based on density functional theory (DFT), which allowed us assignation of bands.

## Results

**Electrochemical performance of PAQS in Li and Mg systems.** PAQS is a well-performing redox-active polymer with a

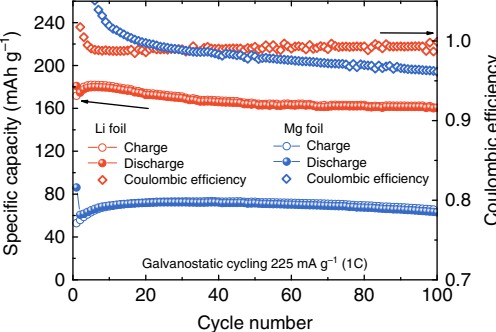

**Fig. 1** Electrochemical performance of Li– and Mg–PAQS batteries. Specific capacity (left y-axis) and Coulombic efficiency (right y-axis) for PAQS in Li (red) and Mg (blue) systems

theoretical capacity of 225 mAh g⁻¹ per mass of active polymer, which is independent on the used metal anode. It has already been used in several metal–organic battery systems[8,14,16,23]. In our experiments Li–PAQS battery shows good electrochemical activity with discharge capacity of 180 mAh g⁻¹ in the first cycle at 1C current density (Fig. 1), which is in good agreement with previously published results[8]. Slow capacity decrease and stable Coulombic efficiency above 99% of PAQS-active material suggests reversible mechanism of electrochemical storage. Good electrochemical performance was also observed in Mg system, where Coulombic efficiency stayed above 96% throughout the cycling. In starting cycles, it was above 100% due to gradual decrease of polarization in initial cycles (Supplementary Fig. 1) and irreversible reactions of electrolyte on Printex carbon at low potentials (Supplementary Fig. 2)[24], but then it slowly declined down to 96% in the 100th cycle (Fig. 1). Discharge capacity in Mg system reached 72 mAh g⁻¹ after 25 cycles due to gradual activation of the electrode composite. Afterwards only small capacity decay was observed, and discharge capacity in the 100th cycle was 63 mAh g⁻¹. Lower capacity in Mg system was mainly attributed to poorer accessibility of active material in Mg battery system and gradual activation of Mg anode[25]. Electrochemical cycling at lower current densities decreases differences between capacities in Li and Mg systems (Supplementary Fig. 3). Nevertheless, both systems exhibited long-term cycling stability at 1C with well-defined electrochemical plateaus (Supplementary Fig. 1a-b).

**In operando ATR-IR mechanism investigation in Li system.** Excellent reversibility of PAQS in both systems suggests a reversible mechanism, which does not depend on the used cation. A hypothetical, but experimentally never confirmed, mechanism for the electrochemical activity of PAQS in a Li–organic battery was proposed by Song et al.[8]. (Fig. 2a). To explore the mechanism of PAQS, we designed a pouch cell with a Si wafer window on the cathode side (Supplementary Fig. 4). Si wafer window allowed us continuous measurement of ATR-IR spectra during electrochemical cycling. PAQS showed high electrochemical activity with a characteristic monotonous change of the voltage between 2.5 and 2.0 V vs. Li/Li⁺ (Fig. 2b). A discharge capacity of 154 mAh g⁻¹ was obtained in the first cycle and gradually increased in the second and third cycles up to 168 mAh g⁻¹, which is in the agreement with cycling in the Swagelok cell configuration (Fig. 1). During the first cycle a polarization of 178 mV was measured in the middle of the discharge curve, which decreased down to 79 mV in the third cycle.

PAQS has several characteristic IR bands: the C=O stretching vibrations are located at 1676 and 1651 cm⁻¹ and the aromatic ring −C=C− bond stretching vibrations at 1570 cm⁻¹, which is an area where the electrolyte and Si wafer peaks do not interfere

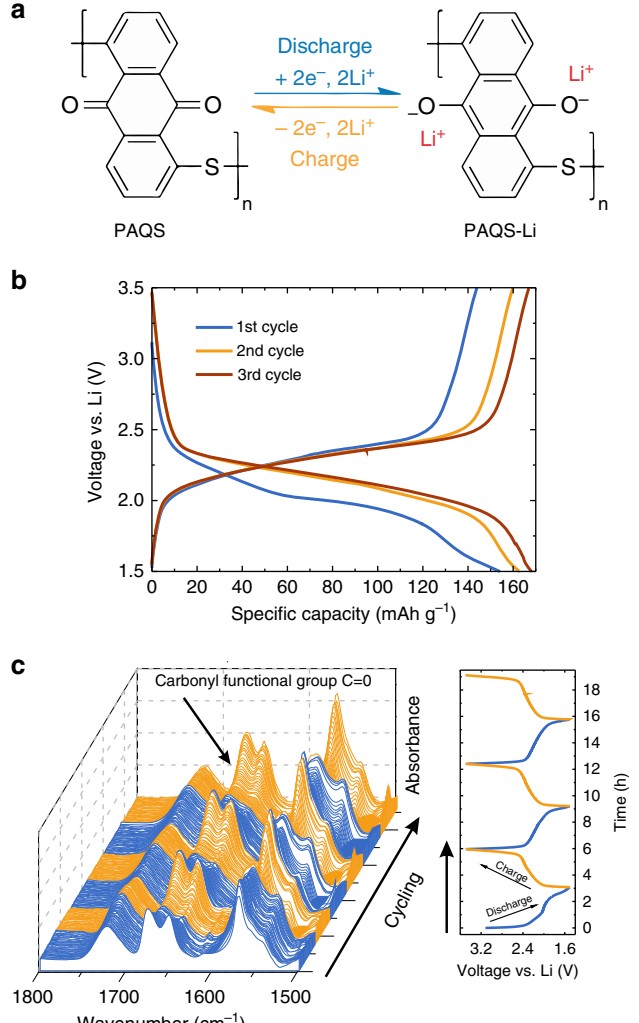

**Fig. 2** Characterization of the Li–PAQS electrochemical mechanism through in operando ATR-IR. **a** Proposed electrochemical mechanism of PAQS in a Li–organic battery. **b** Discharge/charge cycles for the 1st, 2nd and 3th cycles of PAQS in the modified pouch cell with a Si wafer window at 50 mA g$^{-1}$ current density. **c** Corresponding ATR-IR spectra of the PAQS cathode during galvanostatic cycling in the region from 1800 to 1500 cm$^{-1}$. Blue and orange spectra correspond to discharge and charge, respectively

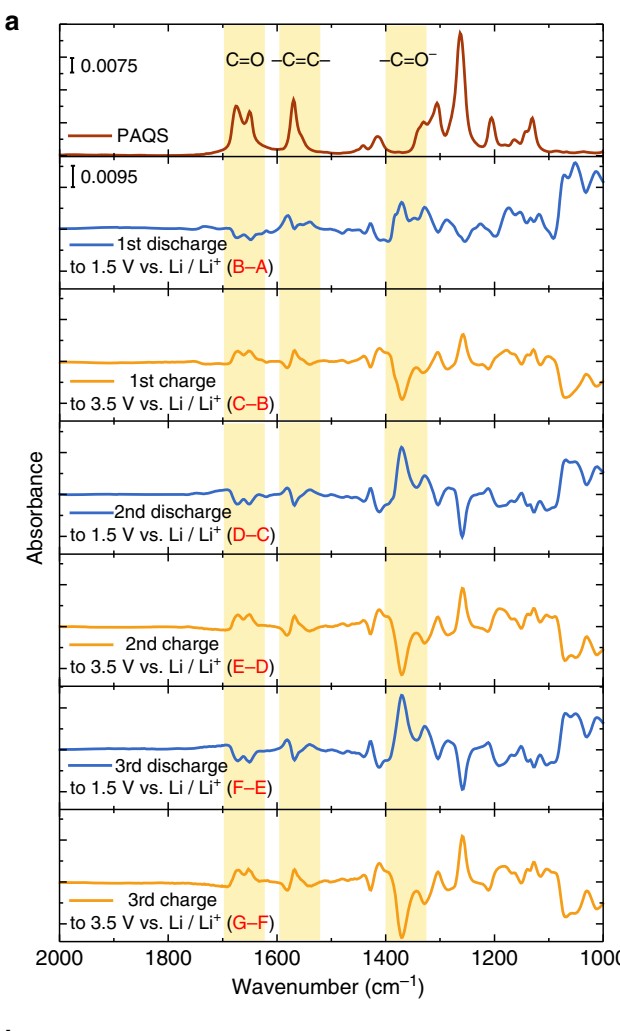

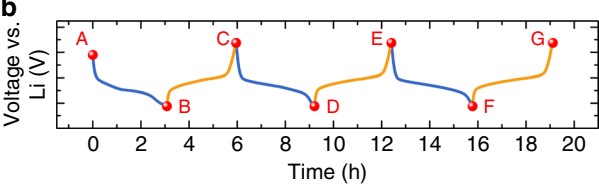

**Fig. 3** The ATR difference spectra. **a** Spectrum of the PAQS and difference spectra of the PAQS cathode during cycling. The absorbance scale is identical for all difference spectra. **b** The first three cycles for PAQS cathode vs. time with the marked points from where the subtracted spectra were taken

(Supplementary Fig. 5). Figure 2c shows time-dependent evolution of the spectra during the first three cycles. The intensity of C=O bands gradually decreased during discharging and increased during charging. The same pattern was observed for the aromatic ring stretching at 1570 cm$^{-1}$. Increase and decrease of the intensity of the C=O vibrations confirms high reversibility of the reduction of carbonyl bonds.

The changes observed in the spectra could be caused by alterations of other cathode components during cycling (the carbon black, binder and electrolyte). To test this hypothesis, a cathode composite made out of only Printex XE2 carbon black and polytetrafluoroethylene (PTFE) binder (in a mass ratio of 70:30) was characterized. Galvanostatic cycling gave a typical double-layer capacitive response with 32 mAh g$^{-1}$ of capacity (Supplementary Fig. 6). The ATR-IR spectra did not show any changes during cycling (Supplementary Fig. 7). The main IR bands of these spectra were attributed to the Si wafer window and the electrolyte (Supplementary Fig. 5). This measurement indicates that the changes observed in the spectra of the PAQS cathode are caused by changes of the active material.

The ATR spectra of the PAQS cathode recorded during cycling (Fig. 2c) revealed information about the electrochemical mechanism from the part of the spectra where no background of the electrochemical cell and the electrolyte are present (Supplementary Fig. 5). More information can be gained by subtracting two spectra recorded at the end or at the beginning of the reduction or oxidation process from each other. With that, all vibrational bands, which are unaffected by electrochemical cycling, can be eliminated (mainly electrolyte, Si wafer) (Fig. 3). Thus, we were able to observe even small spectral changes inside the cathode composite that were caused by electrochemical cycling. The ATR difference spectra (Fig. 3a) were obtained from in operando spectra (Fig. 2c) recorded close to the cut-off voltages during the discharge/charge cycles (Fig. 3b). The spectrum B−A from Fig. 3a was obtained by subtraction of the spectrum recorded at the

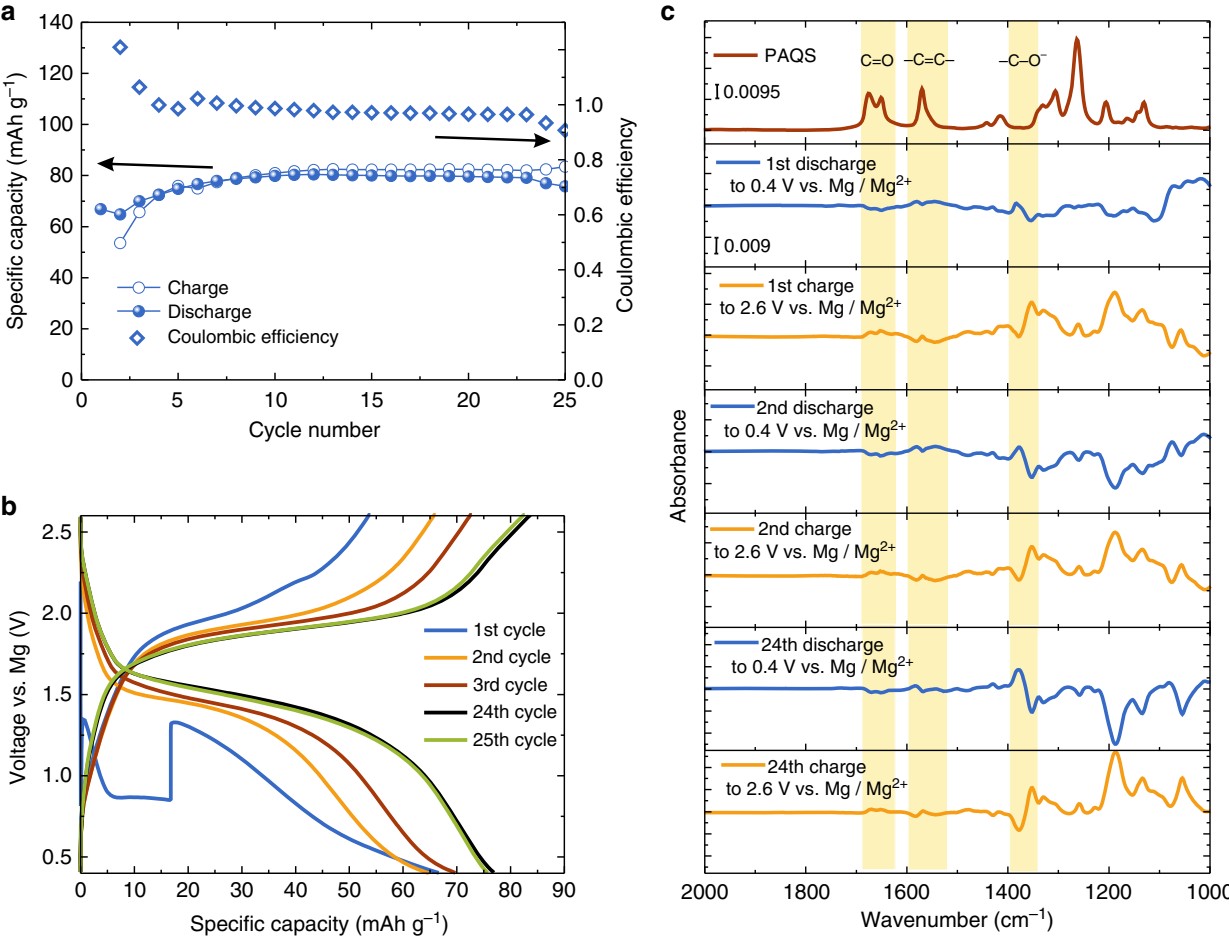

**Fig. 4** Electrochemical performance of Mg–PAQS battery and the ATR difference spectra. **a** Specific capacity (left y-axis) and Coulombic efficiency (right y-axis) with cycle number for PAQS in Mg–organic battery. **b** Selected galvanostatic discharge/charge cycles for the same battery. **c** Selected difference spectra for the PAQS cathode during cycling in the Mg battery

initial state (point A, Fig. 3b) from the spectrum recorded at the end of the first discharge (point B, Fig. 3b). The calculated ATR difference spectra displayed negative or positive ATR absorbance differences in accordance with the decrease or increase in intensity of a particular band during a specific cycling step, respectively (Fig. 3a).

The difference spectra confirmed the observed decrease in intensity of the C=O and −C=C− bands and visualized additional IR-active changes that were previously concealed by peaks from the cell and electrolyte. A strong new band was revealed at 1370 cm$^{-1}$ that increased upon discharging. In the first discharge, this band was rather weak, which we attribute to the formational nature of the first discharge. In subsequent cycles it became the most prominent band in the difference spectra. This band belongs to the stretching of the −C−O$^{-}$ Li$^{+}$ bond that is present in the discharged form of PAQS. The band at 1370 cm$^{-1}$ was accompanied by a less intense band at 1427 cm$^{-1}$, which exhibited the same behavior during cycling. A strong −C−O$^{-}$ Li$^{+}$ band is also observed in the synthesized Li salt of Li$_2$AQ (Supplementary Fig. 8), which represents the discharged form of AQ in the Li battery system. This removes any possible ambiguities arising from the literature that assign −C−O$^{-}$ Li$^{+}$ band to values between 1100 and 1000 cm$^{-1}$ [21]. Furthermore, upon discharge of PAQS we also observed a strong increase in the intensity for a broad double peak at 1070 and 1050 cm$^{-1}$, which we attributed to new ring vibrations (C−C stretching and in-plane C−H bending). This observation agreed with Li$_2$AQ which has ring vibration bands at

1070 and 1025 cm$^{-1}$ (Supplementary Fig. 8). Difference spectra showed the reversible decrease and increase of bands at 1306 and 1263 cm$^{-1}$. The changes in ring vibrations can be attributed to the change in the aromaticity between the neutral (PAQS) and its ionized form (PAQS$^{2-}$).

An attempt was made to confirm and characterize the structural changes during electrochemical characterization of the redox-active polymers with ex situ XRD. Unfortunately, poorly crystalline PAQS and Li–PAQS did not give us any exact information about the electrochemical mechanism (Supplementary Note 1, Supplementary Figure 9).

To confirm the applicability of our in operando ATR-IR method to other compounds, we extended our study to PAQ (Supplementary Fig. 10a) and PANI (Supplementary Fig. 11a) electroactive compounds[9,26]. The PAQ electrochemical reduction and oxidation exhibit the same mechanism as observed in PAQS, proving electrochemical activity of carbonyl group. PANI is a redox-active conjugated polymer with nitrogen atoms incorporated in the polymer backbone, which allow it to possess a range of oxidation states. Upon electrochemical cycling, it shows the most changes for aromatic ring stretching in the region between 1600 and 1500 cm$^{-1}$. Furthermore, the incorporation of solvent molecules is observed, which is in the agreement with previous reports obtained by electrochemical quartz crystal microbalance[26] and with IR studies using PANI model compounds[27] (see the Supplementary Notes 2 and 3).

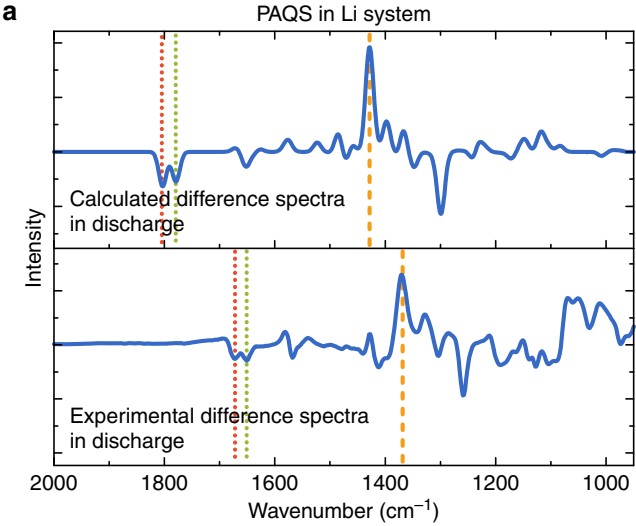

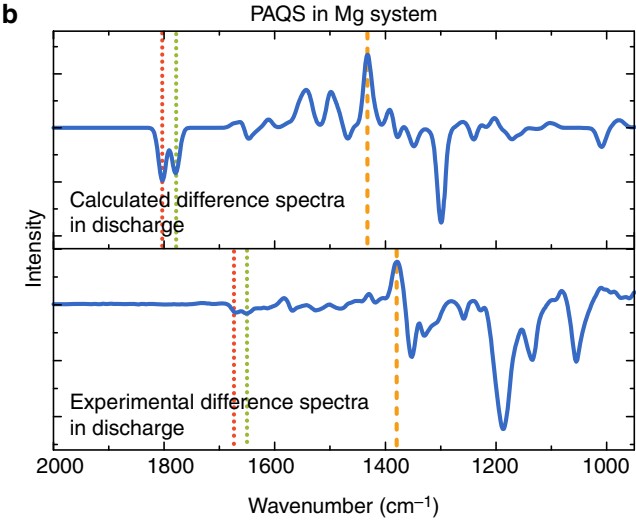

**Fig. 5** Comparison of the calculated and experimental difference spectra in discharge with assigned vibrational modes. **a** Calculated and experimental difference spectra in discharge for PAQS in Li battery system. **b** Calculated and experimental difference spectra in discharge for PAQS in Mg battery system. Dotted lines represent C=O vibrational modes of PAQS and dashed lines represent C–O⁻ vibrational modes of PAQS²⁻

| **Table 1 Comparison of experimental and theoretically calculated carbon-oxygen bond frequencies in PAQS and PAQS²⁻ as well as AQ and AQ²⁻** | | |
|---|---|---|
| **Wavenumber (cm⁻¹) of C=O and C–O stretching**[a] | | |
| **System** | **Experiment** | **Theory** |
| PAQS | 1670, 1650 | 1803, 1777 |
| PAQS²⁻ (Li system)[b] | 1370 | 1426 |
| PAQS²⁻ (Mg system)[b] | 1376 | 1432 |
| AQ | 1678 | 1814 |
| AQ²⁻ | 1370 | 1510 |

[a]The multiplicity of the C–O bond differs between neutral (PAQS) and ionized form (PAQS²⁻)
[b]C–O stretchings in PAQS²⁻ are coupled with ring vibrations, and the vibration with highest C–O⁻ contribution is reported

mechanism is reversible through continuous reduction and oxidation of carbonyl bond, regardless of the divalent cation involved into the reaction. However, the slight upshift for the C–O⁻ band from 1370 cm⁻¹ in Li system to 1376 cm⁻¹ in Mg system is related to the interaction of the carbonyl bond with Mg²⁺ ions rather than Li⁺. Pronounced changes between the difference spectra of PAQS in the Mg and Li battery systems are observed in the fingerprint region. PAQS peaks at 1306 and 1263 cm⁻¹ (ring vibrations of C–C stretching and in-plane C–H bending) are not retained in the charged state. Instead, a new peak at 1187 cm⁻¹ is observed in the charged state that reversibly disappeared upon discharge. The changes in ring vibration modes point to structural changes in PAQS. These changes are due to the interaction with Mg²⁺ ions, which cause conformational changes of PAQS polymer. Moreover, a good reversibility is observed even after more than 20 cycles as can be seen from the difference spectra of the 24th cycle (Fig. 4c). The reason for poorer performance of PAQS in Mg compared to Li battery system is due to much larger solvation shell of Mg²⁺ which hinders accessibility of Mg²⁺ ions to the bulk of the polymer[29]. Unfortunately, lower capacities in the Mg battery mean that the signals are not as high as in the Li battery. However, elucidation of the PAQS electrochemical mechanism was still possible and is an important step towards improved performance of Mg-organic batteries.

Experimentally observed changes in the IR spectra which were assigned to structural changes of PAQS due to interactions of reduced phase with Li⁺ and Mg²⁺ ions were assessed by DFT calculations.

**In operando ATR-IR mechanism investigation in Mg system.** With an aim to probe the mechanism in Mg battery and to check applicability of our method for post Li batteries we characterized electrochemical reaction mechanism of PAQS material coupled with magnesium metal anode. Discharge capacities of 66.9, 64.9 and 70.0 mAh g⁻¹ were obtained in the first three cycles, respectively (Fig. 4a). In the first discharge we observed an activation in the form of a big voltage step (Fig. 4b), which is typical for the breakdown of the passive film on the Mg anode[28]. After this activation, the capacity gradually increased and reached discharge capacities of 77.0 and 75.8 mAh g⁻¹ in 24th and 25th cycle, respectively (Fig. 4a, b), thus exemplifying good capacity stability.

Difference spectra for PAQS in the Mg battery showed similar changes as those in the Li battery. Both exhibited a decrease in the intensity of C=O stretching bands and −C=C− stretching upon discharge and the appearance of a new band at 1376 cm⁻¹, which belongs to −C–O⁻ Mg²⁺ (Fig. 4c). This is in agreement with results obtained in Li system, i.e., electrochemical storage

**DFT calculations.** All of the experimental observations were confirmed by calculation of harmonic frequencies for both PAQS and PAQS²⁻ entities in Li and Mg systems. The calculated spectra at charge and discharge points as well as the difference spectra from the discharged states are shown in Supplementary Fig. 12. Comparison of theoretical and experimental difference spectra enabled assignment of the vibrational modes to experimental vibrational bands (Fig. 5). We can see that good qualitative agreement is achieved. Experimentally and theoretically obtained IR frequencies of the C=O and C–O⁻ stretching modes for PAQS and PAQS²⁻ in Li and Mg systems are presented in Table 1 along with C=O and C–O⁻ stretching modes for reference compound AQ and AQ²⁻. All other bands are attributed to ring vibrations that include C–C stretching and in-plane C–H bending.

Experimentally observed and theoretically calculated shifts can be explained with the concentration of negative charge around the oxygen atoms in PAQS²⁻ which causes a change in the character of the C=O bond from double in the charged state to single bond (C–O⁻) in the discharged state. This manifests itself as 7.6 and 8.3% change in the C–O bond length in Li and Mg

system, respectively. The C–O bond elongates from 1.22 Å for C=O to 1.313 Å (1.321 Å) for C–O$^-$ in Li (Mg) system, while the difference in average C–H and C–C bond lengths is negligible (Supplementary Table 1). Thus, the calculated IR frequencies of C–O$^-$ stretching in the discharged state shift to lower values relative to the frequencies of C=O in the charged state (Fig. 5, Table 1). This is in accordance with experimental data, as the experimental difference spectrum shows a decrease in the C=O bands at 1676 and 1651 cm$^{-1}$ and the appearance of bands at approximately 1427 and 1370 cm$^{-1}$ (attributed to C–O$^-$) (Fig. 5). Theoretical calculations were validated by comparing calculated IR spectra of model compounds AQ and AQ$^{2-}$ with measured spectra of AQ and lithium salt (Li$_2$AQ). (Table 1, Supplementary Fig. 8 and Supplementary Fig. 13).

## Discussion

In this work we studied the electrochemical mechanism of a PAQS cathode in Li and Mg battery systems. Our study was performed by application of newly developed in operando ATR-IR in a modified pouch cell with a Si wafer window. Simultaneous measurements of IR spectra during cycling revealed monotonous decrease and increase of the intensity of carbonyl bond. More detailed information from the measured spectra was obtained by subtraction that removed interference peaks from the Si wafer and electrolyte. That allows visualization of all IR-active changes occurring inside the cathode composite. The ATR-IR measurements on PAQS were supported by measurements of the reference compound AQ and its synthesized reduced counterpart Li$_2$AQ and complemented with DFT calculations. This represents an important step towards improving our understanding of batteries with organic electrodes. Furthermore, we were able to observe pronounced variations between the difference spectra of PAQS in Li and Mg batteries for ring vibrations, which are connected to differences in the polarizing power between Mg$^{2+}$ and Li$^+$ ions. Nevertheless, after initial changes Mg battery system is stable in latter cycles as it can be seen from in operando measurements.

Finally, we extended the scope of our method to PAQ and PANI to demonstrate its wider applicability. Our designed analytical method can serve as a straightforward tool for studies of different research problems in the battery research and broader electrochemical community. Besides being a powerful analytical tool for determination of mechanism in the organic batteries, it can also serve as a tool for monitoring degradation processes during the electrochemical characterization of materials or study of the electrolyte stability.

## Methods

**Synthesis**. Poly(anthraquinonyl sulfide) PAQS was synthesized as described in the literature[8] by polycondensation of 1,5-dichloroanthraquinone with Na$_2$S in N-methyl-2-pyrrolidone and characterized by IR.

PAQ was synthesized according to the literature[9]. The obtained yellow product (89 % yield) was not pure according to $^1$H NMR spectra and was further purified by repetitive washing with MeOH using Soxhlet extractor for 1 day. Purified product had $^1$H NMR and IR spectra according to the literature[30].

Polyaniline (emeraldine base): 200 mL of 1 M HCl and 5 mL of aniline was put into a 500 ml round-bottom flask. The solution was cooled down to −10 °C. Afterwards, 50 mL of 0.2 M ammonium persulfate was added dropwise. The reaction mixture was stirred for 3 h and subsequently filtered, washed with water, ethanol and acetone. The intermediate product was in the next step dispersed for 20 min into a 200 mL of 1 M ammonium hydroxide solution and stirred for 1 h. The obtained emeraldine base was finally filtered and washed with water, ethanol and acetone. The final product was dried at 50 °C under vacuum.

9,10-Dihydroxy anthracene (HAQ): Anthraquinone (208 mg, 1 mmol) was dissolved in DMSO (10 mL). Sodium borohydride (40 mg, 1 mmol) was added to the above solution and the color of solution changed to dark red. Reaction mixture was stirred for 6 h at room temperature. Afterwards it was poured into 1 M HCl (50 mL) to obtain precipitate, which was subsequently filtered, washed with water two times and finally dried under vacuum (50 °C, overnight) to get fluorescent

yellow-green powder (159 mg, yield 76%). All procedures including storage of the sample were performed under inert Argon atmosphere inside the glovebox to prevent oxidation. IR (KBr, inert atmosphere): 3317 (br), 1620, 1389, 1331, 1234, 1170, 1138, 1052, 759, 627 cm$^{-1}$, $^1$H NMR (300 MHz, DMSO-$d_6$) δ 9.41 (s, 1H), 8.34 (dd, $J$ = 6.8, 3.2 Hz, 2H), 7.41 (dd, $J$ = 6.8, 3.2 Hz, 2H). $^1$H NMR spectrum is in agreement with the literature[31].

Lithium salt of 9,10-dihydroxy anthracene (Li$_2$AQ): 9,10-dihydroxy anthracene (52.5 mg, 0.25 mmol) was dissolved in anhydrous MeOH (7.5 mL) at 50 °C. To this solution 2 M LiOMe in MeOH (0.25 mL, 0.5 mmol, 2 eq) was slowly added drop by drop and color of solution changed from fluorescent yellow-green to dark violet-red. Reaction mixture was stirred for another 5 h at 50 °C then solvent was evaporated under reduced pressure to obtain dark red-brown powder (56 mg, yield 101%). All procedures including storage were performed under inert Ar atmosphere inside the glovebox to prevent oxidation. Even under those conditions sample quickly degraded in a few days at room temperature—lower storage temperature is recommended. IR* (KBr discs, nujol suspension, inert atmosphere): 1369, 1070 cm$^{-1}$, $^1$H NMR* (300 MHz, DMSO-$d_6$) δ 8.56 (s, 1H), 8.47 (d, $J$ = 9.2 Hz, 2H), 7.78 (d, $J$ = 8.7 Hz, 2H), 7.29–7.18 (m, 2H), 6.91–6.78 (m, 2H). (*NMR spectrum indicates partially protonated species. In both IR and NMR spectra other lower peaks were observed and cannot be fully avoided due to fast degradation of the product.)

**Electrochemical characterization**. Cathode composites were prepared by mixing PAQS, Printex XE2 carbon black and PTFE binder in a mass ratio of 60:30:10. The composite was pressed on a carbon-coated Al mesh at 5 t cm$^{-2}$. Li foil (FMC, thickness 500 μm) and 1 M LiTFSI in dimethoxyethane/1,3-dioxolane (DME/DOL) (1:1 vol%) were used as an anode and electrolyte, respectively. For the Mg battery test, Mg foil (Gallium Source, 99.95%, 0.05 mm) and 0.4 M Mg(TFSI)$_2$ 0.4 M MgCl$_2$ in tetraglyme/1,3-dioxolane (TEG/DOL) (1:1 vol%) were used as an anode and electrolyte, respectively. The anode and the cathode were separated with a GF/A Whatman glass fiber separator. Conventional battery tests were carried out in Swagelok battery cells in a potential window from 1.5 to 3.5 V vs. Li/Li$^+$ or from 0.4 V to 2.6 vs. Mg/Mg$^{2+}$ with a current density of 225 mA g$^{-1}$ (1C rate). In operando battery tests were carried out in a modified pouch cell using a Bio Logic SP200 potentiostat/galvanostat in the same potential windows with a current density of 50 mA g$^{-1}$ (roughly C/5 rate). A Si wafer window was attached to the cathode side of the pouch cell to allow in operando ATR-IR measurements of the cathode composite (Supplementary Fig. 4).

**In operando ATR-FTIR cell**. The spectroelectrochemical in operando ATR-FTIR cell was prepared as a two electrode pouch cell with an IR-transmissive silicon wafer window (Silicon Quest International) and with aluminum and nickel contacts (Supplementary Fig. 4a). The silicon wafer had a crystal orientation of (100) with a thickness of 490 to 510 μm. During the in operando spectroelectrochemical measurements the silicon wafer window on the cathode side of the modified pouch cell was pressed on the ATR Ge crystal to allow measurement of the IR spectra (Supplementary Fig. 4b).

**IR characterization**. ATR-IR measurements were performed on a Bruker IFS-66/2 with a liquid nitrogen cooled mercury cadmium telluride (MCT) detector and equipped with a Specac Silver gate ATR with a Ge crystal. The ATR-IR spectra were collected in absorbance mode with 64 scans at a resolution of 4 cm$^{-1}$ in the range of 4000–500 cm$^{-1}$. Spectra were collected in operando with a series of consecutive scans every 3 min during galvanostatic cycling with current density of 50 mA g$^{-1}$ (roughly C/5 rate). The automatic baseline correction and atmospheric compensation were performed on the ATR-IR spectra in OPUS version 7.0 software. ATR difference spectra of discharge/charge were obtained by subtracting the spectrum of the previous discharge/charge state from the current discharge/charge spectrum.

**Computational details**. All calculations were performed by Gaussian 09 software[32]. Following full geometry optimization, harmonic vibrational frequencies and IR intensities were computed by the M06-2X hybrid functional developed by Zhao and Truhlar[33], and the 6-31+G(d,p) basis set. A polymeric structure of PAQS was modeled with three monomeric units to ensure reasonable accuracy of the resulting vibrational modes while keeping the computational costs manageable. The influence of the surrounding medium was mimicked by a model of a dielectric continuum (polarizable continuum model) with a dielectric constant of 7.4. The computed normal modes were used as a qualitative tool for comparison and analysis of both experimental and theoretical spectra. Vibrational modes with contributions from the methyl groups arise from the finite size of the model and are not included in the analysis (Supplementary Fig. 14). It should also be noted that the presently employed theoretical model tends to overestimate the frequencies of bands of interest (see Supplementary Table 1 and Fig. 5 in the main text) by about 5%. Such discrepancies are commonly observed in quantum calculations and can be attributed to several approximations and simplifications of the model, among the rest the treatment of surroundings, the electronic Hamiltonian and the harmonic approximation. These discrepancies are often minimized by applying a scaling factor and rigid shifts by a certain amount of wavenumbers to obtain optimal agreement with the experimental data. We emphasize that we did not use

any scaling factors or rigid shifts, as this would not improve quality of results on a fundamental level. Nevertheless, the qualitative trend of presented theoretical data is correct and the computed frequency shifts are in good agreement with experimental observations.

**Data availability**. The data supporting the findings of this study are available from the corresponding author on reasonable request.

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

## Acknowledgements

We gratefully acknowledge Professor Dr. Miran Gaberšček for useful discussion, Helena Spreizer for advice and technical help during the in operando ATR-IR measurements, Nejc Pavlin for providing polyaniline and Ana Robba for the XRD measurements. The authors acknowledge the financial support from the Slovenian Research Agency (research project J2-8167 and research core funding Nos. P2-0393, P1-0010 and P1-0012) and Honda R&D Europe (Germany).

## Author contributions

A.V. designed the in operando cell, performed the IR experiment and characterization. J. B. carried out the electrochemical experiments. K.P. synthesized the organic polymer material and the model compounds. A.K.L. performed the DFT calculations. A.V. carried out data processing and prepared the figures. J.B. wrote the manuscript. J.G., J.S., A.R.-V. and R.D. provided input with the data analysis, helped with the discussion and assisted with the manuscript correction. All authors have given approval to the final version of the manuscript.

## Additional information

**Competing interests:** The authors declare no competing financial interests.

