## [Peer Review File · Nature Communications]

Reviewers' comments:

Reviewer #1 (Remarks to the Author):

In this manuscript, the authors reported a poly(antraquinoyl sulfide) based Li-ion and Mg-ion batteries and investigated the electrochemical reaction mechanism by in operando ATR-IR approach. The new in-situ IR technology had been successfully demonstrated by the author to study the IR active changes of poly(antraquinoyl sulfide) inside the electrode composite. These results supported that in both Li and Mg battery systems reduction of carbonyl bond was demonstrated as a mechanism of electrochemical activity, which was also consistent with the DFT calculations. However, the study on the poly(antraquinoyl sulfide) as organic cathode material for Li and Mg batteries demonstrates no significant novelty and advance, because poly(antraquinoyl sulfide) (PAQS) has been well-explored in several battery systems (Li, Na, Mg, K) in previous reports.^{7,13,15,22} In addition, it is well known that carbonyl bond of poly(antraquinoyl sulfide) (PAQS) is always suggested as the electrically active functionality. In this reports, no significant new mechanism insights has been revealed. The wide scope application of this analytical tool for determination of mechanism in organic batteries has not been well-demonstrated, as the author also tested one carbonyl compound. The quality of the paper could be improved if the author could further demonstrated that the in operando ATR-IR approach can be really serves as a universal analytic tool for determination of mechanism in many types of organic electrode materials such as different types of carbonyl compounds, poly imines, organodisulfides, quinoxaline-like molecules etc. Therefore, although this report is a technically sound in mechanism study of organic batteries, it is not qualified for the high standards of this Journal.

Other minor issues:

1. In this manuscript, "Electrochemical activity of PAQS was first tested in Li battery system, where discharge capacity of 180 mAh g⁻¹ was obtained in the first cycle (Figure 1a) followed by gradual capacity decrease of 20 mAh g⁻¹ in 100 cycles." Actually, Ref.7 has reported the PAQS for LIBs in 2009.
2. The electrochemical performances of PAQS for Li and Mg batteries should compare with the published work. Ref. 7 and 13.
3. The author should explain why the initial discharge curve in Mg battery suddenly increased?
4. Since the authors studied both Li and Mg batteries in this work, and the differences and comparison of these two batteries should be stated clearly.
5. The figures in the manuscript should be well-rearranged. The charge/discharge curves are plotted and put in different figures. i.e. figure 1b and figure 2b are presenting the repeated information. Figure 1c and Figure 5b are same. Therefore the figure 1 is suggested to SI part.
6. At present, the peak intensities of the in-situ results are quite low in this work. The advantages of this in-situ technique should be proved in comparison to Ex-situ IR results or another characterizing techniques.
7. Some important papers on reaction mechanism study for organic electrodes should be cited.

The manuscript could be revised and then submitted to other journal such as Scientific Reports.

Reviewer #2 (Remarks to the Author):

I read with interest the manuscript of Vizintin and co-workers. The study looks at the reactions of Li and Mg with an organic cathode, i.e. poly(antraquinoyl sulphide), utilizing a combination of electrochemistry, in situ Infrared spectroscopy and first-principles calculations. Please find my review in the attachment.

Reviewer #3 (Remarks to the Author):

I think this is an interesting article trying to understand the what is the mechanism behind the charge/discharge of organic cathode PAQS for both Li/Mg batteries.

The authors choose a pouch cell as the experimental device and ATR-IR as the spectroscopic tool. The mechanism has been indicated by several authors already (Ref 7 and Ref 14) and the results provide an experiment evidence which agrees with the previous publication that the C=O group underwent redox to (C-O)⁻anion.

It would be nice that there is some extra analytical tool to confirm the results, for example, in-situ XRD etc, however, the tool ATR-IR is quite accessible and I believe this manuscript can provide help for the community to use this technique.

Overall, this is a good article to read and see and will help for development of more in operando tools to study the battery system.

Response to Reviewers

We have carefully considered all the comments and questions raised by the reviewers. We took time to plan and carry out additional experiments, which helped to address some of the reviewers' comments and validate wider applicability of *in operando* ATR-IR method. Newly obtained data are included in the manuscript and supplementary information. The majority of the changes in the manuscript are related to the discussion about the wide applicability of the *in operando* ATR-IR method. We extended our method to polyanthraquinone (PAQ) and polyaniline electroactive compounds with a concise description of the results in main text and detailed discussion of the experiments with additional figures in the supplementary information (Figure S9-S10). Additionally, we have improved and extended the DFT calculations. In the DFT model we have explicitly included Li^+ and Mg^{2+} cations in the discharged model of PAQS. These two new DFT models, now better represent active material in the discharged state. Furthermore, the model with explicit Mg^{2+} shows conformational changes that were not observed in previous model. As pointed out by the reviewers the original version of the manuscript contained some claims and related conclusions that were not justified in a proper manner. Thus, we clarified some of the statements in the manuscript with the support of the new experimental data and additional discussion. We kindly thank the reviewers for exposing relevant questions and constructive comments which, in our opinion, helped us to improve the quality and clarity of the present work.

Here are point by point answers on the reviewers' comments with corresponding changes in the revised version of the manuscript.

Reviewers' comments:

Reviewer #1 (Remarks to the Author):

In this manuscript, the authors reported a poly(antraquinonyl sulfide) based Li-ion and Mg-ion batteries and investigated the electrochemical reaction mechanism by *in operando* ATR-IR approach. The new *in-situ* IR technology had been successfully demonstrated by the author to study the IR active changes of poly(antraquinonyl sulfide) inside the electrode composite. These results supported that in both Li and Mg battery systems reduction of carbonyl bond was demonstrated as a mechanism of electrochemical activity, which was also consistent with the DFT calculations. However, the study on the poly(antraquinonyl sulfide) as organic cathode material for Li and Mg batteries demonstrates no significant novelty and advance, because poly(antraquinonyl sulfide) (PAQS) has been well-explored in several battery systems (Li, Na, Mg, K) in previous reports.^{7,13,15,22} In addition, it is well known that carbonyl bond of poly(antraquinonyl sulfide) (PAQS) is always suggested as the electrically active functionality. In this reports, no significant new mechanism insights has been revealed. The

wide scope application of this analytical tool for determination of mechanism in organic batteries has not been well-demonstrated, as the author also tested one carbonyl compound. The quality of the paper could be improved if the author could further demonstrated that the *in operando* ATR-IR approach can be really serves as a universal analytic tool for determination of mechanism in many types of organic electrode materials such as different types of carbonyl compounds, poly imines, organodisulfides, quinoxaline-like molecules etc. Therefore, although this report is a technically sound in mechanism study of organic batteries, it is not qualified for the high standards of this Journal.

Answer:

We thank the reviewer for the comments and suggestions; however we disagree in some parts.

The electrochemical mechanism of organic electroactive compounds are most often based on the IR measurements of the *ex-situ* samples. The *ex-situ* samples, especially the discharged compounds, are extremely unstable and prone to degradation. Especially, during sample transfer out of the glove box to the IR spectrometer, where they can come in contact with moisture/air, and also with time as observed for Li₂AQ in our work. Moreover, with *ex-situ* measurements we are risking experimental errors and bias due to the fact that different electrodes have to be used for charged and discharged electrode measurements. To overcome above mentioned difficulties, a real-time monitoring or *in operando* measurements of the reaction mechanism involved in the battery is required. While PAQS has been used in several battery systems (Li, Na, Mg, K), electrochemical mechanism of PAQS has so far never been investigated. Therefore, its mechanism is highly interesting, especially when moving from monovalent (Li) to bivalent (Mg) battery system. During the battery operation, with the *in operando* ATR-IR measurements one can follow in real-time the disappearance of the carbonyl peaks of PAQS (1676 and 1651 cm⁻¹) and appearance of new bands, especially the –C–O⁻ (1370 cm⁻¹ in case of Li and 1376 cm⁻¹ in case of Mg). This removes any possible ambiguities arising from the literature that assign –C–O⁻ Li⁺ band to values between 1100 and 1000 cm⁻¹.¹ For the reasons stated above, we think, that *in operando* ATR-IR analytical method provides novel results through real-time insight into PAQS cathode material during the battery operation in a Li and Mg system. Implementation of this method in the organic battery research field will ease electrochemical mechanism investigation and help with the design of future cathode materials.

We thank the reviewer for the suggestions on how to improve the quality of our manuscript. For this reason we have extended the wide scope application of this *in operando* ATR-IR analytical tool for mechanism determination in organic batteries to polyanthraquinone (PAQ) and polyaniline electroactive compounds. **Figures R1** and **R2** show our new *operando* ATR data on two additional compounds, which confirm the wide applicability of *in operando* ATR-IR method. A concise

description of the results has been included in the manuscript, while in depth discussion of the results was included in the supplementary file.

The PAQ was cycled at 50 mAh g^{-1} in the voltage window of 1.5 V to 3.0 V inside the *in operando* ATR-IR cell. Capacities around 219 mAh g^{-1} were achieved. Similar electrochemical mechanism as in the case of PAQS can be expected for PAQ (**Figure R1a**). PAQ exhibited high electrochemical activity with a characteristic monotonous change of the voltage between 2.5 and 2.0 V (**Figure R1b**). Similar changes in the voltage were seen in PAQS. In the IR spectrum (**Figure R1c, red curve**), PAQ has two characteristic peaks located at 1670 and 1592 cm^{-1} . The two peaks are assigned to the stretching vibrations of the C=O and $-\text{C}=\text{C}-$ bond, respectively.² During the discharge, a decrease of intensity for C=O and $-\text{C}=\text{C}-$ stretching bands and appearance of new $-\text{C}-\text{O}^- \text{Li}^+$ band (1373 cm^{-1}), can be seen in the differential spectrum. In charge the situation is reversed, bands for C=O and $-\text{C}=\text{C}-$ stretching increase in intensity, while band for $-\text{C}-\text{O}^- \text{Li}^+$ decreases (**Figure R1c**).

Figure R1. a) A proposed electrochemical mechanism for polyanthaquinone (PAQ) in lithium battery system. a) Galvanostatic cycling in voltage range from 1.5 to 3.0 V vs. Li/Li^+ in the *in operando* ATR-IR in lithium battery cell. b) ATR IR spectrum of PAQ (brown), differential spectrum of first discharge (blue) and differential spectrum of first charge (yellow).

Polyaniline (emeraldine base) in **Figure R2a** was cycled at 50 mAh g^{-1} in the voltage range from 2.5 (except the first discharge, which was to 2.0 V to achieve lithiation of emeraldine base) to 4.0 V Li/Li^+ (**Figure R2b**). Capacities around 137 mAh g^{-1} were achieved. In the IR spectrum (**Figure R2c, red curve**) of the polyaniline (emeraldine base) characteristic peaks are observed. The polyaniline (emeraldine base) has two strong vibrational peaks for the aromatic ring stretching $-\text{C}=\text{C}-$ at 1593 and 1502 cm^{-1} . Furthermore, the polyaniline has peaks for $-\text{C}-\text{H}$ bending and $-\text{C}-\text{N}$ stretching at 1309 , 1167 and 1109 cm^{-1} .^{3,4} During the cycling of polyaniline, changes in the ATR spectra are observed. In

the ATR difference spectrum (**Figure R2c**), upon discharge we can see increase of the bands characteristic for EC:DEC (1803, 1776, 1741, 1259, 1159 and 1072 cm^{-1}), consistent with the introduction of solvent in polyaniline, and upshift of the bands between 1500 and 1600 cm^{-1} , which is characteristic for changed ratio between phenyl and quinoid rings in favour of phenyl ones. Upon charge the bands connected with the carbonate solvents decrease and the bands in the region between 1500 and 1600 cm^{-1} downshift, which is characteristic for increased amount of quinoid rings (**Figure R2c**). This observations are in agreement with the recent study made on polyaniline, where cycling behaviour was investigated using electrochemical quartz micro balance,³ and previous *ex-situ* IR study on model compounds of polyaniline.⁴

Figure R2. a) Polyaniline (emeraldine base) structure and proposed electrochemical reaction mechanism. b) Galvanostatic cycling in the voltage range from 2.5 (except first cycle to 2.0 V to achieve lithiation of emeraldine base) to 4.0 V Li/Li^+ in the *in operando* ATR-IR in lithium battery cell. c) ATR-IR spectrum of polyaniline-emeraldine base (brown), EC:DEC solvent (green), differential spectrum of second discharge (blue) and differential spectrum of second charge (yellow).

Other minor issues:

1. In this manuscript, “Electrochemical activity of PAQS was first tested in Li battery system, where discharge capacity of 180 mAh g⁻¹ was obtained in the first cycle (Figure 1a) followed by gradual capacity decrease of 20 mAh g⁻¹ in 100 cycles.” Actually, Ref.7 has reported the PAQS for LIBs in 2009.

Answer:

To make the sentence more clear and remove ambiguity in the main manuscript, we have changed it into:

In our experiments Li-PAQS battery shows good electrochemical activity with discharge capacity of 180 mAh g⁻¹ in the first cycle at 1 C (**Figure 1**), which is in good agreement with previously published results⁸.

2. The electrochemical performances of PAQS for Li and Mg batteries should compare with the published work. Ref. 7 and 13.

Answer:

The electrochemical performance of PAQS in Li battery system is in a good agreement with Ref 7 at similar current densities. PAQS in Mg battery system exhibits lower capacity in starting cycles, but significantly better long-term cycling stability. However, different electrolyte and anode have to be taken into account. Please note that Ref. 7 and Ref. 13 are in new version of manuscript Ref. 8 and Ref. 14.

We changed the text in following sentences.

In our experiments Li-PAQS battery shows good electrochemical activity with discharge capacity of 180 mAh g⁻¹ in the first cycle at 1 C (**Figure 1**), which is in good agreement with previously published results⁸.

Discharge capacities in initial cycles are lower as in our previous work¹⁴, but Mg-PAQS battery shows significantly better long-term cycling stability. However, different electrolyte and anode have to be taken into account.

3. The author should explain why the initial discharge curve in Mg battery suddenly increased?

Answer:

In the first discharge in Mg-PAQS battery, the sudden increased in the voltage in a form of activation is typical for the breakdown of the passive film on the Mg foil anode.⁵ The sentence is present in the main manuscript in the section of mechanism investigation through *in operando* ATR-IR in Mg system.

4. Since the authors studied both Li and Mg batteries in this work, and the differences and comparison of these two batteries should be stated clearly.

Answer:

We thank the reviewer for this remark. The electrochemistry of Li and Mg batteries is now compared in the first paragraph of the text. Detailed investigation of mechanism is now described in two separated paragraphs; Mechanism investigation through in operando ATR-IR in Li system and Mechanism investigation through in operando ATR-IR in Mg system. Furthermore, in the section DFT calculation of the new models for Li and Mg system, the detailed description of both models allows comparison between the mechanisms in these two systems through theoretical calculations.

5. The figures in the manuscript should be well-rearranged. The charge/discharge curves are plotted and put in different figures. i.e. figure 1b and figure 2b are presenting the repeated information. Figure 1c and Figure 5b are same. Therefore the figure 1 is suggested to SI part.

Answer:

We moved the previous Figure 1, the galvanostatic charge/discharge curves to the supplementary information.

6. At present, the peak intensities of the in-situ results are quite low in this work. The advantages of this in-situ technique should be proved in comparison to Ex-situ IR results or another characterizing techniques.

Answer:

It is true that the intensities for C=O and for the changes are relatively low in the *in operando* ATR-IR measurements. This is due to the high refraction index of Ge bearing crystal and relatively high absorption index of Si window used on the pouch cell to allow measurement of the IR spectra of the cathode during electrochemical characterization. Additionally relatively high amount of carbon black (30 %) and binder in the electrode (10%) that was used in the electrode, lowers the intensities of PAQS and electrolyte peaks in the battery cell (**figure R3**). The main contributions in the IR spectra from the *in operando* measurement are from the Si wafer (1721, 1448, 1301, 1233, 1108, 959, 890, 817, 738 and 689 cm^{-1}) and electrolyte (1457, 1355, 1335, 1191, 1137, 1083, 1062 cm^{-1}). However, the comparison of pure PAQS and the PAQS electrode in the *in operando* cell with the electrolyte (**figure R3**) shows that despite of lower intensities of C=O and –C=C– stretching vibration peaks

bands are still clearly observable. As mentioned before *ex-situ* measurements can introduce experimental errors and bias due to the fact that different electrodes are used for each measurement. Presence of electrolyte in the electrode attributes additional IR bands that interfere the interpretation of the electrode spectra, while excessive washing of electrodes to remove electrolyte can partially dissolve the active material or passive layers that are created upon cycling. On other hand, calculation of difference spectra efficiently removes the Si and electrolyte IR bands and allows visualization of even small changes in band intensity and/or frequency of active substance(s) upon electrochemical cycling. Most importantly *in operando* ATR IR avoids decomposition of samples, which are often highly sensitive to air and moisture. Additionally, it allows relatively long-term monitoring of the battery activity up to few tens of cycles as shown in this work.

Figure R3: The ATR-IR spectra of pure PAQS (black) and the PAQS in the cathode inside the operando cell with the electrolyte (red)

7. Some important papers on reaction mechanism study for organic electrodes should be cited.

Answer:

Several papers were cited in the original manuscript and revised version. Specifically, Ref. 17 where *ex-situ* XRD and IR are used for determination of reduction mechanism of lithium terephthalate and lithium muconate. In Ref. 18 *ex-situ* electrodes are used for ¹³C SS NMR for mechanism investigation of poly(2,5-dihydroxy-1,4-benzoquinonyl sulphide). In Ref. 19 *ex-situ* IR and *in-situ* Raman are used for investigation and confirmation of mechanism of 2,5-dihydroxyterephthalic acid. In Ref. 20 electrochemical mechanism of polyquinoneimines was investigated with *ex-situ* IR spectroscopy. Similarly in Ref. 21, *ex-situ* IR spectroscopy was used for investigation of sodium anthraquinone-1,5-disulfonate. To the best of our knowledge these are some of the most important papers, in which electrochemical mechanism of organic electrodes has been studied.

Reviewer #2 (Remarks to the Author):

I read with interest the manuscript of Vizintin and co-workers. The study looks at the reactions of Li and Mg with an organic cathode, i.e. poly(antraquinoyl sulphide), utilizing a combination of electrochemistry, in situ Infrared spectroscopy and first-principles calculations. The authors developed a new in operando IR cell/setup to monitor the conversion reactions occurring in the cathode upon reaction Li(Mg). Upon Li(Mg) discharge, IR measurements detected the disappearance of C=O signatures to form new C–O– modes, which are nicely reproduced by Density Functional Theory calculations. Furthermore, the in situ IR measurements attested the reversibility of the conversion process of the poly(antraquinoyl sulfide). Although the manuscript is relevant for the battery community and meeting the novelty standards of Nature Communications, the paper needs to integrate some changes as elaborated below.

We thank the reviewer for the supportive review and careful examination of the manuscript. His insightful comments helped to expose areas, where we were able to improve our work.

Comments:

1. The authors do perform IR in situ characterization to assess Li(Mg) reactivity with the cathode. While I am convinced that reversible conversion reactions of the poly(antraquinoyl sulphide) takes place, could the author supplement the in situ IR measurement with some additional XRD (in situ or ex situ) data? In my opinion the validity of the claims can only be rationalized if the observation derived from IR are also confirmed with complementary techniques. Additional characterizations will show that the conversion reactions claimed by the authors are the only electrochemical process occurring in the electrochemical cell.

Answer:

Anthraquinone (AQ) is a crystalline solid that has monoclinic crystal structure, due to the π - π stacking of the anthracene rings (**Figure R4, blue diffractogram**).⁶ On the other hand, poly(anthraquinoyl sulfide) (PAQS) exhibits poor crystallinity with low peak intensities as shown in the **Figure R4 (red diffractogram)**. These peaks are connected with presence of short chain oligomers of PAQS and not AQ monomers.⁶ While it is possible to determine exact structure of AQ from XRD and observe structural changes upon cycling, this is in our opinion impossible in case of PAQS and other poorly crystalline or amorphous compounds. This emphasises the importance of *in operando* IR spectroscopy, which is one of the few methods that allow visualization of molecular changes inside PAQS and other organic compounds.

Figure R4: XRD pattern of AQ and PAQS.

2. The revised manuscript should state what is the theoretical gravimetric capacities for both Li and Mg reactions. These are crucial for comparison with what is measured by the electrochemical tests.

Answer:

We have included the information about theoretical gravimetric capacity in the following sentence.

PAQS is a well performing redox active polymer with a theoretical capacity of 225 mAh g^{-1} that has already been used in several metal-organic battery systems.^{8,14,16,23}

3. The statement “Lower capacity observed in Mg battery system is typically affected by worse accessibility of electrolyte to active sites of electroactive polymers” is vague and does not fully explain the electrochemical data (Figure 5a and b). If one assume that on Mg will react with two C=O groups simultaneously ensuing one electron per C=O unit, the theoretical gravimetric capacity should be close to that of Li. The 2 discrepancy between the theoretical gravimetric capacity and the experimental observation should be elaborated better in the text. In addition, the author should show in the SI Current vs. Potential plots to show whether there are overpotentials upon Li(Mg) discharge.

Answer:

To demonstrate the effect of poorer accessibility of the electrolyte to active sites of the polymer we have tested both batteries at lower current density (22.5 mA g^{-1} or $C/10$). As we have shown in the manuscript the Mg-PAQS at 1C exhibits an initial discharge capacity of around 80 mAh g^{-1} , which faded down to 72 mAh g^{-1} after 25 cycles. At a current density of $C/10$ an initial discharge capacity of 235 mAh g^{-1} was reached. The slightly higher achieved first discharge capacity compared to

theoretical one (225 mAh g^{-1}) is due to the high surface area Printex carbon black additive (the 30 wt.% of Printex inside the cathode will give up to 41 mAh g^{-1} of extra capacity in the first discharge (see **Figure R6** in the answer 5)). In the initial 5 cycles capacity is decreases to 120 mAh g^{-1} and after 34 cycles 134 mAh g^{-1} discharge capacity is achieved. This shows that if we cycle at lower current densities differences between capacities in Li and Mg system are smaller and also activation process in Mg system is less pronounced.

Figure R5: Discharge capacity for PAQS in Li (red) and Mg (blue) systems at 22.5 mA g^{-1} (C/10) current density.

The decrease of the over-potential in Mg system can be seen in **figure 4b** in the main text and in the supplementary information (**figure S1b**). In both figures decrease of the over-potential is especially pronounced in the initial cycles.

4. In the results section the author stated, “High coulombic efficiency was also observed in Mg system where coulombic efficiency stayed above 96% throughout cycling”. I consider High Coulombic efficiency anything above 99 % and not below. As such, I suggest the authors to tone down this claim.

Answer to 4:

We thank the reviewer for the suggestion. The claim was toned down and the sentence was modified into:

Good electrochemical performance was also observed in Mg system where Coulombic efficiency stayed above 96% throughout the cycling.

5. The manuscript often reports vague claims/conclusions which are not fully justified by the data:

a. In the result section: “In starting cycles, (the coulombic efficiency) it was above 100% due to the gradual activation connected with some irreversible reactions...” The authors claim the existence of irreversible reactions, what are those? Are these reactions occurring at the anode or the cathode? In general, it has been shown that Mg(TFSI)₂ in glymes is likely to decompose at the Mg anode. This is never discussed in the manuscript. In addition, the manuscript should at least cite some of the scientific work published on this subject, i.e., J. Am. Chem. Soc. 2015, 137, 3411–3420 DOI: 10.1021/jacs.5b01004, ACS Energy Lett., 2017, 2 (1), 224–229, DOI: 10.1021/acsenergylett.6b00549, J. Phys. Chem. C 2016, 120, 3583–3594, DOI: 10.1021/acs.jpcc.5b08999, J. Phys. Chem. Lett. 2016, 7, 1736–1749, DOI: 10.1021/acs.jpcllett.6b00384, Chem. Rev. 2014, 114, 11683–11720, DOI: 10.1021/cr500049y.

b. Another not fully explained remark, “Lower capacity in Mg battery system was mainly attributed to poorer electrolyte accessibility due to different solvent used in Mg battery system and gradual activation of the anode”. This statement is vague and may need some references (see point a.)

c. The sentence “These changes are due to interaction with Mg²⁺ ions, which polarize PAQS more strongly than Li⁺ ions, however more detailed study needs to be performed in order to clearly elucidate changes during formational cycles.” is vague and does not help. It is well known that Mg is a more polarizing anion than Li, but the authors do not endeavour to explain how this can affect the structure of PAQS. Please remove this sentence in the revised manuscript.

d. The statement “The reason for poorer performance of PAQS in Mg battery is due to much larger solvation shell of Mg²⁺ which hinders accessibility of Mg²⁺ ions to the bulk of the polymer.” should be justified by some literature, perhaps Phys. Chem. Chem. Phys., 2014,16, 21941-21945 DOI: 10.1039/c4cp03015j .

Answer:

Point a): To confirm the presence of the irreversible reactions a battery with Printex carbon black (70 % Printex and 30 % PTFE binder) as cathode and Mg foil as anode was tested at equivalent of C/10 current density. In the cycling of Printex cathode certain degree of irreversible reactions can be observed, especially in the lower voltage region in the starting cycles. The irreversible reactions most likely occur on the both, anode and cathode, but only those on cathode affect the Coulombic efficiency, since Mg foil anode is present in a large surplus over cathode. In the cycling of Printex cathode irreversible reactions occur at lower voltage limits in discharge and cause Coulombic efficiency above 100 %. We believe that this side reactions are caused by decomposition of the electrolyte as mentioned in the references. This effect also translates into the battery cycling and is additionally accompanied with decrease of polarization of the battery, which also raises the Coulombic efficiency in initial cycles.

We have restructured the sentence in the paragraph and added some of the recommended references. Furthermore, we added in the supporting information the galvanostatic cycling of the Printex cathode.

Figure R6 Galvanostatic discharge/charge curves for the cathode made out of Printex XE2 carbon black and PTFE binder in Mg system with 0.4 M Mg(TFSI)₂ 0.4 M MgCl₂ in TEG:DOL electrolyte at 11.3 mA g⁻¹ (C/10) current density.

For the point b): “Lower capacity in Mg battery system was mainly attributed to poorer electrolyte accessibility due to different solvent used in Mg battery system and gradual activation of the anode” see the answer to the comment # 3.

For point c): The sentence is changed in the revised version into:

These changes are due to interaction with Mg²⁺ ions, which cause conformational changes of PAQS polymer. Influence of Li⁺ and Mg²⁺ ions on PAQS structure and IR spectra will be investigated in the following section with the help of DFT calculations.

For point d): We have added the reference to justify the sentence.

6. The authors seem to dismiss some important literature, especially on the topic of Mg batteries. In general, I request the authors to add at least the seminal work on Mg batteries by D. Aurbach, Nature 2000, 407, 724–727, DOI: 10.1038/35037553 and the recent review Chem. Rev. 2017, 117, 4287–4341, DOI: 10.1021/acs.chemrev.6b00614 .

Answer:

We thank the reviewer for pointing out that by focusing on organic batteries we have missed two important literature papers on the Mg batteries. We added both references.

7. Figure 3a, I believe the last number in the x-axis should be 1000 cm^{-1} and not 100. Also, the authors provide yellow shades to highlight relevant areas in the IR spectra; please add labels to each shading of the assigned IR modes. The same should be applied to Figure 5c.

Answer:

We thank the reviewer for noticing the mistake on the x-axis. We corrected it and added the labels to each yellow shading.

8. Figure 4, the dark green curve is never explained both in the caption and in the text.

Answer:

The dark green curve represented the theoretical spectrum of PAQS²⁻, which was used for calculation of theoretical difference spectrum in the old version of manuscript. In the revised version only theoretical difference spectrum is displayed in the main manuscript, while other theoretical spectra are moved into the supplementary information.

9. In the description of the frequency shifts that the IR modes undergoes upon Li(Mg) discharge/charge, I suggest using the terminology red-shift and blue-shift when appropriate.

Answer:

Red-shift and blue shift are very common expressions in the IR community, but they are used for shifts of the specific band that undergoes a blue- or red-shift. In our work the C=O band disappears and new C-O⁻ band appears. Thus, we think that use of such terminology could be a bit misleading for certain readers. No changes in the manuscript.

10. The Harmonic frequencies computed through DFT were obtained on the PAQS and PAQS²⁻ monomers, surprisingly the authors did not consider explicitly the interaction of Li (or Mg) with PAQS²⁻. This may be the origin of the substantial discrepancy of the shifts between experiment and theory. The revised manuscript should include at least a model and its frequencies in which PAQS²⁻ is coordinated by 2 Lithium or 1 Mg.

Answer:

We thank the reviewer for this important remark. We have now modelled PAQS²⁻ with Li and Mg explicitly added to the system. Relaxed geometry of a polymeric structure of PAQS, Li-PAQS and Mg-PAQS, modelled with three monomeric units and terminated with methyl group can be seen in **figure R7**.

Figure R7: Relaxed geometry of a polymeric structure of PAQS, Li-PAQS and Mg-PAQS, modelled with three monomeric units and terminated with methyl group.

Consideration of explicit interaction of Mg and Li with PAQS²⁻ has improved and expanded the information delivered by calculations, especially on the Mg system. However, the new models have not significantly reduced the discrepancy between experimental and calculated peak frequencies.

These kind of discrepancies are commonly observed in quantum calculations and are attributed to several approximations and simplifications of the model, among the rest the treatment of surroundings, the electronic Hamiltonian, and the harmonic approximation. These discrepancies are often reduced by applying a scaling factor, which is obtained empirically from comparison of experimental and theoretical vibrational frequencies. Often, additional rigid shifts by a certain amount of wavenumbers are applied to obtain optimal agreement with the experimental data. Theoretical model that we employed in this work overestimates the frequencies of bands of interest by about 5%, which is in accordance with precomputed vibrational scaling factor for the M06-2X/6-31+G(d,p) level of theory that we used in this work^{7,8}. Comparison between experimental and theoretical results with and without applied scaling factor can be seen in **Figure R8** and **Figure R9**.

Figure R8. Comparison of experimental (bottom) and theoretical difference spectrum (top) in Li system without any scaling factor (top graph, dashed red line) and with applied scaling factor of 0.95 (top graph, blue solid line).

Figure R9. Comparison of experimental (bottom) and theoretical difference spectrum (top) in Mg system without any scaling factor (top graph, dashed red line) and with applied scaling factor of 0.95 (top graph, blue solid line). Rigid shift of 50 wavenumbers additionally improved agreement with the experimental spectra.

Although application of precomputed vibrational scaling factor for the level of theory that we use in this work considerably improves visual agreement of theoretical and experimental results, it does not improve quality of results on a fundamental level and has no solid physical meaning. Furthermore, qualitative trend of the computed frequencies is correct and the computed frequency shifts are in good agreement with experimental observations with or without application of a scaling factor. We emphasise that for these reasons we have not used scaling factor in previous submission and we have also decided not to use it in the current work.

If the editor or reviewers insist, that application of the scaling factor contributes to the quality of the presented work, e.g., arguing that scaling factors are routinely used in some of the related research articles in this field, we will include scaling factor in our theoretical results and analysis.

11. The authors should also re-compute the vibrational frequencies with a higher quality basis-set than a 6-31+G (d,p), for example, cc-pVTZ or AUG-cc-pVTZ with the addition of polarization and diffuse functions.

Answer to 11:

First, we would like to emphasise that we used three monomeric units to model PAQS, rather than a monomer, as suggested by the reviewer in his comment 10. The enlarged model significantly increases the needed computational time. We used hybrid functional M062X of Truhlar and Zhao⁹ which includes dispersion correction and is considered one of best functionals available, evolving into a gold standard in quantum chemistry. We used the 6-31+G(d,p) basis set¹⁰ which included diffuse and polarization functions. On this level of theory, the number of basis functions for three monomeric units was 1162. If we increased the basis set to aug-cc-pVTZ as suggested by the reviewer, the number of basis functions increased to 3052, which renders the calculation highly impractical, if not even impossible for our resources. To illustrate the exorbitant cost of such calculation, the estimated computational time required to complete one step of geometry optimization is about three days on a 20-core intel Xeon E5-2660 v3 server with 64 GB RAM, expecting the optimization (typically tens of steps) to be completed in several months, yet not mentioning the even more demanding harmonic frequency calculation. Nevertheless, we tested the performance of the suggested basis set to a model consisting of only one monomeric unit, which allowed us to perform optimization and compute the frequencies. These frequencies were critically compared with the ones obtained by the original basis set and/or larger models. The comparison is displayed in **figure R8**. It is evident that the change of the basis set does not significantly change the resulting frequencies, whereas changing the model from one monomeric unit to three monomeric units gives results that are in much better accordance with the experiment. Based on this arguments, we decided to keep the 6-31+G(d,p) basis set and three monomeric units as a PAQS model.

Figure R10: Comparison of the impact of the basis set and number of the monomeric units in the PAQS model.

Top: theoretical spectra of PAQS. Red: results obtained with aug-cc-pVTZ basis set, PAQS is modelled with one monomeric unit; Green: results obtained with 6-31+G(d,p) basis set, PAQS is modelled with one monomeric unit; Blue: results obtained with 6-31+G(d,p) basis set, PAQS is modelled with three monomeric units;

Bottom: experimental result of PAQS.

The above comparison shows that enlargement of the basis set does not improve the results significantly, while it drastically increases the computational cost. In contrast, expansion of the PAQS model from one monomeric unit to three monomeric units has a substantial impact on the results in that it significantly improves the agreement with experimental spectra. Based on this results we keep the 6-31+G(d,p) basis set and model PAQS with three monomeric units.

12. In the intro, please replace “Quantum calculations” with “Quantum Mechanical calculation”.

Answer:

The quantum mechanical calculation was added to the introduction of the manuscript.

13. In the intro, the author should justify better how organic materials are so convenient compared to traditional inorganic transition metal based cathodes. I’m not really convinced organic materials represent a turning point in terms of both energy density and costs. Please tone down these sentences.

Answer:

We thank the reviewer for the comment. We have changed the sentence to:

Among these technologies metal-organic batteries attract increasing attention due to their versatility, low-cost and sustainability.

Reviewer #3 (Remarks to the Author):

I think this is an interesting article trying to understand the what is the mechanism behind the charge/discharge of organic cathode PAQS for both Li/Mg batteries.

The authors choose a pouch cell as the experimental device and ATR-IR as the spectroscopic tool. The mechanism has been indicated by several authors already (Ref 7 and Ref 14) and the results provide an experiment evidence which agrees with the previous publication that the C=O group underwent redox to (C-O)⁻anion.

It would be nice that there is some extra analytical tool to confirm the results, for example, in-situ XRD etc, however, the tool ATR-IR is quite accessible and I believe this manuscript can provide help for the community to use this technique.

Overall, this is a good article to read and see and will help for development of more in operando tools to study the battery system.

We thank the reviewer for the supportive review.

Answer:

Anthraquinone (AQ) is a crystalline solid that has monoclinic crystal structure, due to the π - π stacking of the anthracene rings (**Figure R4, blue diffractogram**). On the other hand, poly(anthraquinoyl sulfide) (PAQS) exhibits poor crystallinity with low peak intensities as shown in the **Figure R4 (red diffractogram)**, most likely these peaks are connected with presence of short chain oligomers of PAQS and not AQ monomers.⁶ While it is possible to determine exact structure of AQ from XRD and observe structural changes upon cycling, this is impossible in case of PAQS and other poorly crystalline or amorphous compounds. This demonstrates the importance of *in operando* IR spectroscopy, which is one of the few methods that allow visualization of molecular changes inside PAQS and other organic compounds.

Figure R4: XRD pattern of AQ and PAQS.

We tried performing a complementary *in-situ* Raman. Unfortunately, the PAQS is unstable under the Raman laser (1064 nm, 785 nm and 532 nm). The presence of fluorescence by applying a 532 nm laser and extreme heating of the sample caused by lasers that emit at 1064 nm and 785 nm significantly reduce the applicability of Raman spectroscopy. Furthermore, in Raman spectrum the polar groups, such as C=O and C–O⁻ have relatively low intensity in comparison to corresponding bands in IR spectrum.

Therefore, we have performed the DFT calculations on a Li and Mg PAQS models to confirm the observation from the *operando* ATR.

References

1. Wan, W. *et al.* Tuning the electrochemical performances of anthraquinone organic cathode materials for Li-ion batteries through the sulfonic sodium functional group. *RSC Adv.* **4**, 19878 (2014).
2. Song, Z. *et al.* Polyanthraquinone as a Reliable Organic Electrode for Stable and Fast Lithium Storage. *Angew. Chemie Int. Ed.* **54**, 13947–13951 (2015).
3. Jiménez, P. *et al.* Lithium n-Doped Polyaniline as a High-Performance Electroactive Material for Rechargeable Batteries. *Angew. Chemie - Int. Ed.* **56**, 1553–1556 (2017).
4. Boyer, M.-I. *et al.* Vibrational Analysis of Polyaniline: A Model Compound Approach. *J. Phys. Chem. B* **102**, 7382–7392 (1998).
5. Tutusaus, O., Mohtadi, R., Singh, N., Arthur, T. S. & Mizuno, F. Study of Electrochemical Phenomena Observed at the Mg Metal/Electrolyte Interface. *ACS Energy Lett.* **2**, 224–229 (2017).
6. Song, Z., Zhan, H. & Zhou, Y. Anthraquinone based polymer as high performance cathode material for rechargeable lithium batteries. *Chem. Commun.* 448–450 (2009).
7. Peverati, R. & Truhlar, D. G. Quest for a universal density functional: the accuracy of density functionals across a broad spectrum of databases in chemistry and physics. *Philos. Trans. R. Soc. A Math. Phys. Eng. Sci.* **372**, 20120476–20120476 (2014).
8. Johnson, R. D. I. CCCBDB Computational Chemistry Comparison and Benchmark Database. *CCCBDB Computational Chemistry Comparison and Benchmark Database* (1999).
9. Zhao, Y. & Truhlar, D. G. The M06 suite of density functionals for main group thermochemistry, thermochemical kinetics, noncovalent interactions, excited states, and transition elements: two new functionals and systematic testing of four M06-class functionals and 12 other function. *Theor. Chem. Acc.* **120**, 215–241 (2008).
10. Ditchfield, R., Hehre, W. J. & Pople, J. A. Self-Consistent Molecular-Orbital Methods. IX. An Extended Gaussian-Type Basis for Molecular-Orbital Studies of Organic Molecules. *J. Chem. Phys.* **54**, 724–728 (1971).

REVIEWERS' COMMENTS:

Reviewer #1 (Remarks to the Author):

This manuscript after some revisions demonstrates the organic cathodes studied by in operando infrared spectroscopy with impressive analytical technique and reliable mechanism study. The wide applicability of in operando IR technique is very interesting. We suggest this manuscript accepted by Nature Communications after some minor revisions.

1. The authors should move all the experimental results in the Response letter to the Supporting Information part.
2. The authors did not answer the Question 2 and 3 of from the previous reviewer 2 very well. That is very important to the theoretical capacity of Mg.
3. It is better to show both charge and discharge capacities in Figure 1.
4. If possible, some ex-situ XRD results of AQ during electrochemical process are necessary to confirm the reliability of the in-situ IR results.

Response to Reviewers

We thank the reviewer for the comments and suggestions. We have carefully considered all points raised by the reviewer. Some additional *ex-situ* XRD measurements were carried out, which show loss of long range periodicity in the discharged state of PAQS polymer, which is reflected as a disappearance of the diffraction peak at 24° . Although this experiment shows changes in the structure of PAQS polymer, it does not provide an exact information about the molecular mechanism of the electrochemical reaction. This emphasises the importance of *in-operando* IR spectroscopy, which is one of the few methods that allow visualization of molecular changes inside PAQS and other organic compounds.

Here are point by point answers on the reviewer comments with the corresponding changes.

Reviewer #1 (Remarks to the Author):

This manuscript after some revisions demonstrates the organic cathodes studied by in operando infrared spectroscopy with impressive analytical technique and reliable mechanism study. The wide applicability of in operando IR technique is very interesting. We suggest this manuscript accepted by Nature Communications after some minor revisions.

1. The authors should move all the experimental results in the Response letter to the Supporting Information part.

All the related experimental results from the response letter are moved to the Supporting Information part. Supporting information part additionally includes full explanation of the *in-operando* IR measurements on PAQ and polyaniline, the DFT calculation on the model compounds and the XRD measurements on AQ, PAQS and *ex-situ* XRD on the PAQS cathodes.

2. The authors did not answer the Question 2 and 3 of from the previous reviewer 2 very well. That is very important to the theoretical capacity of Mg.

The Question 2 from the previous reviewer 2 was regarding the theoretical gravimetric capacities for both Li and Mg reaction. In both systems the cathode is made out of the same material, e.g. PAQS, thus the theoretical capacity in both systems is the same (theoretical capacity for PAQS is 225 mAh g^{-1}).

We modified the sentence in the manuscript to explain this better, which is now the followed:

„PAQS is a well performing redox active polymer with a theoretical capacity of 225 mAh g^{-1} , which is independent on the metal anode used. It has already been used in several metal-organic battery systems.^{8,14,16,23**}

Regarding the question 3, connected with poorer accessibility of the electrolyte to the active sites of PAQS, we conducted additional experiments where PAQS was galvanostatically discharged and charged with lower current density. The obtained results are shown in the

supporting information, supplementary Figure 3. Electrochemical characterisation performed with lower current density shows substantial increase of capacity for both studied systems. This effect is especially pronounced for the Mg system, where discharge capacity increases from 69 mAh g⁻¹ in 10th cycle at 1C (225 mA g⁻¹) to 118 mAh g⁻¹ in 10th cycle at 0.1C (22.5 mA g⁻¹). This suggests that the lower capacity for the Mg system is predominantly due to ionic wiring problems. Lower capacity in the formation cycles can be ascribed partially to the polarisation caused by anode as can be seen in the Figure 4b in the main text and supplementary Figure 1b.

3. It is better to show both charge and discharge capacities in Figure 1.

We thank the reviewer for this remark. The Figure 1 is now modified and both charge and discharge capacities are shown besides the Coulombic efficiency.

4. If possible, some ex-situ XRD results of AQ during electrochemical process are necessary to confirm the reliability of the in-situ IR results.

The following paragraph about the XRD measurements was added to the main text of the manuscript:

„ An attempt was made to confirm and characterize the structural changes during electrochemical characterisation of the redox active polymers with ex-situ XRD. Unfortunately, poorly crystalline PAQS and PAQS2- did not give us any exact information about the electrochemical mechanism (Supplementary Note 1, Supplementary Figure 9).“

The results of ex-situ XRD are now included into the Supporting information file under the section Supporting notes 1 as described below:

XRD measurements. The powder XRD measurements were carried out and on a Siemens D5000 diffractometer with Cu K α_1 radiation ($\lambda = 1.5406 \text{ \AA}$).

Diffractograms of PAQS and AQ were measured on a PANalytical X'pert PRO high-resolution diffractometer in the range between 10-60° on zero-background holder (Si wafer). The *ex-situ* cathodes were characterized by Siemens D5000 diffractometer with Cu K α_1 radiation ($\lambda = 1.5406 \text{ \AA}$) inside a Swagelok type cell with a 200 μm thick beryllium window in the 2 θ range from 10° to 40°.

Anthraquinone (AQ) is a crystalline solid that has monoclinic crystal structure, due to the π - π stacking of the anthracene rings (Figure R1, green diffractogram).¹ On the other hand, poly(anthraquinoyl sulfide) (PAQS) exhibits poor crystallinity with low peak intensities as shown in the Figure R1 (red diffractogram). These peaks are connected with presence of short chain oligomers of PAQS and not AQ monomers.¹ The PAQS cathodes (fresh (Figure R1, orange diffractogram) and the *ex-situ* cathode in a discharge state (Figure R1, blue diffractogram)) were measured inside a Swagelok type cell with a beryllium window. The

fresh cathode resembles the PAQS structure with a slight shift of the peaks due to the height displacement of the thick Be window. The *ex-situ* cathode in the discharge state exhibits a loss in long range periodicity which is due to formation of a poorly crystalline or amorphous discharged compounds. This emphasises the importance of *in-operando* IR spectroscopy, which is one of the few methods that allow visualization of molecular changes inside PAQS and other organic compounds.

Figure R1 XRD measurements. Synthesised AQ (green diffractogram), PAQS (red diffractogram), fresh PAQS cathode (orange diffractogram) and *ex-situ* PAQS cathode in a discharge state (blue diffractogram). The XRD of the fresh cathode and the *ex-situ* cathode in a discharge state were measured inside a Swagelok type cell with a beryllium window.

Reference:

1. Song, Z., Zhan, H. & Zhou, Y. Anthraquinone based polymer as high performance cathode material for rechargeable lithium batteries. *Chem. Commun.* 448–450 (2009).